# DPP-4 as a Possible Biomarker of Inflammation Before Abdominal Surgery for Chronic Pathology: Our Experience with Elective Cholecystectomy

**DOI:** 10.3390/medicina55050148

**Published:** 2019-05-16

**Authors:** Liliana Valencia-Sánchez, Rafael Almendra-Pegueros, Luis Jose Diaz R-Valdez, David Esmer-Sánchez, Úrsula Medina, Antonio Gordillo-Moscoso

**Affiliations:** 1General Surgery Department, “Dr. Ignacio Morones Prieto” Central Hospital, San Luis Potosí 78290, Mexico; ldv_2001@hotmail.com (L.V.-S.); luisjosediazr@gmail.com (L.J.D.R.-V.); esmer_david@hotmail.com (D.E.-S.); 2Translational Research Laboratory in Pharmacology, Clinical Epidemiology Department, Faculty of Medicine, UASLP, San Luis Potosí 78210, Mexico; ralmendrap@alumnos.uaslp.edu.mx (R.A.-P.); ursula.medina@uaslp.mx (U.M.)

**Keywords:** dipeptidyl peptidase-4, perivesicular inflammation, biomarker, cholecystectomy, pathology, surgical

## Abstract

*Background and objectives:* Dipeptidyl-Peptidase 4 (DPP-4) is a protein expressed in numerous cells and tissues. Recently it has shown its involvement as a catalyst in the inflammatory response in various pulmonary, autoimmune, intestinal and other pathologies. The objective of this study was to compare the preoperative serum levels of DPP-4 in patients with and without surgical finding of perivesicular inflammation. *Materials and methods*: a cross-sectional analytical study nested in a prospective cohort, including patients scheduled for elective cholecystectomy, without surgical complications, that were 18–70 years of age, with low cardiovascular risk, without a history of peritonitis, pancreatitis, or jaundice and underwent ERCP protocol, type 2 diabetes mellitus, acute inflammatory (Protein C Reactive < 3 mg/L, leucocytes < 10 1000/mm^3^), neoplastic, nephrologic or liver disease, the use of anti-inflammatory drugs, steroids and/or antibiotics, the use of pacemakers or metallic implants and without major amputations and whom agreed to participate by providing their informed consent. Ethical and Research register: 45–16. Prior to surgery we compiled anthropometric data and a blood sample to determine the serum levels of DPP-4. The presence of perivesicular inflammation was determined in the surgery. The data was analyzed using the statistical program Rstudio. *Results*: High BMI values were observed (27.8 ± 6.4); waist circumference (94.7 ± 15.1) and percentage of fat mass (34.7 ± 11.7), showing a cumulative frequency of 65.9% for overweight/obesity. In 27.3% of the interventions, intraoperative perivesicular inflammation findings were reported. The serum levels of DPP-4 were lower in the group of patients with perivesicular inflammation (3947.6 ± 1659.5 vs. 3053.2 ± 1469.6, LC95% of the difference: 160.4–1628.3), being statistically significant (*p* = 0.018). *Conclusions*: In the subacute or chronic phases of cholecystitis, there appears to be a constant consumption of DPP-4, which would modulate a better immune response that could be related to the reduction of postoperative complications, so the use of Serum levels of DPP-4 as an early biomarker could improve the diagnostic accuracy of this pathology and the surgical approach.

## 1. Introduction

Dipeptidyl-Peptidase 4 (DPP-4) The CD26 is a trasmembranal protein of 110 kDa, expressed in a wide variety of cells and tissues, including cells of the endothelium, Lymphocytes, Enterocytes, fibroblasts, macrophages, prostate, kidney, pancreas, liver, lungs and adipose tissue [1]. The soluble form of this protein is identified in various biological fluids such as blood plasma, Cerebrospinal fluid, Synovial fluid, bile, saliva, urine and semen [2,3,4,5].

DPP-4/CD26 has important enzymatic activity in the degradation of N-terminal peptides, dipeptides and polypeptides with amino acids such as proline or alanine in the penultimate position. Due to this catalytic activity, DPP-4 regulates the bioavailability of regulatory peptides, neuropeptides, circulating hormones and cytokines [2,5], which are derived from the modulation of multiple biological functions. Of these, the most well documented is the regulation of glucose metabolism from the degradation of GLP-1 (Glucagon-like peptide-1) and GIP (Glucose dependent insulinotropic peptide), resulting in the absence of the insulinotropic effect of these hormones [6]. In addition, other regulated biological processes have been described for DPP-4: carcinogenesis, as a possible tumor suppressor in conjunction with CXCL10 and by post-transcriptional modification of cytokines in the regulation of tumor immunity [2], in addition to being a possible biomarker in the diagnosis of different types of cancer [7,8]; immunity modulation by direct regulation of lymphocytes [2] and in the inflammatory response being mediated by cytokines and the catalyst role of DPP-4, characteristic of chronic inflammatory pulmonary, autoimmune, intestinal and other pathologies.

Somborac-Bačura, in 2012 [9] identified that DPP-4 levels were significantly lower in patients with stable COPD than in healthy controls, suggesting its use as a serological biomarker since this enzyme is not affected by smoking, age and other factors related to pathology. In 2016, Chang et al., [10], confirmed these findings and showed that the serum levels of DPP-4 increased in patients with COPD in remission.

Busso and Collar, in the 2005 [11], identified that in patients with inflammatory rheumatoid arthritis (AR), serum levels of DPP-4 and their enzymatic activity were decreased compared to patients with non-inflammatory AR and osteoarthritis (OA), as well as being inversely correlated with serum C-reactive protein levels (PCR), suggesting that this decrease could influence the regulation of the SDF-1/CXCR4 chemotactic shaft. Results according to this line were described by Sromova et al., 2015 [12], also adding that during the follow up of patients with RA up to six months, during the less active phase of the disease, the levels of DPP-4 increased and this correlated with the decrease in the score of activity of the disease and the decrease of the serum levels of PCR.

In inflammatory bowel disease, such as ulcerative colitis and Crohn’s disease (CD), some inverse correlations have been reported between the enzymatic activity of DPP-4, the clinical activity score of the disease and inflammatory markers (alpha-1-acid glycoprotein) and PCR [13], whereas when analyzing only the population with CD in its active phase, the levels of DPP-4 are decreased in plasma and tissue, suggesting the possible use of this biomarker [14].

The presence of perivesicular inflammation is one of the main indicators of moderate inflammation and is a risk factor for complications in surgical processes. To make an accurate diagnosis, it is necessary to use specialized studies such as abdominal ultrasound which observes air or liquid in the wall of the gallbladder in up to 95% of cases, however, its absence does not exclude the diagnosis and the sensitivity of this diagnostic method is less than the computed axial tomography. However, the case of ultrasound can present false negatives or be confused with porcelain gallbladder or stones attached to the wall. And finally, the realization of these can delay the diagnosis and give rise to a greater appearance of complications [15,16].

Due to the possible advantages of using DPP-4 as a biomarker of inflammatory processes, attempts have been made to identify its serum levels in patients with initial perivesicular inflammation, a pathology that requires diagnostic specialized tools. This is why the objective of this study was to compare the preoperative serum levels of DPP-4 in patients with and without a surgical finding of perivesicular inflammation.

## 2. Materials and Methods

### 2.1. Design of the Study and Selection of Subjects

An analytical cross-sectional study was conducted of patients who did not have postoperative complications in a 30-day follow-up, with a prospective cohort of postsurgical complications in elective cholecystectomy of the Department of General Surgery of the Central Hospital “Dr. Ignacio Morones Prieto” (HCIMP). Complete results from the cohort will be published elsewhere. The cohort includes patients with the following criteria: diagnosis of chronic cholecystectomy 6 month before the study starts, qualified as low cardio-metabolic risk in preoperative assessment, under dietary recommendations in the month prior to surgery, between 18–70 years old and whom agreed to participate by signing informed consent.

Patients with a history of peritonitis, pancreatitis, jaundice and underwent ERCP protocol, type 2 diabetes mellitus, acute inflammatory (Protein C Reactive < 3 mg/L, leucocytes < 10 1000/mm^3^), neoplastic, nephrologic or liver disease, the use of anti-inflammatory drugs, steroids and/or antibiotics, use of pacemakers or metallic implants and with major amputations were excluded. The project was evaluated and accepted by the Research and Ethics Committee of the Central Hospital “Dr. Ignacio Morones Prieto”, obtaining the registration number 45–16 (approved on 30th June 2016).

### 2.2. Procedures

A day before the surgical intervention, a patient interview was conducted to determine its inclusion in the protocol, the signature of informed consent and the registration of anthropometric data.

#### 2.2.1. Anthropometry

We recorded weight and size, to calculate the body mass index (BMI), waist circumference and body fat mass percentage. The weight and percentage of body fat mass were measured with an electric bioimpedance scale TBF-300A Total Body Composition Analyzer (TANITA Corporation, Tokio, Japan), height was measured with stadiometer Seca 201^®^ (Medical Measuring Systems and Scales SECA, Hamburg, Germany) with a limit of 210 cm; And waist circumference with anthropometric tape of glass fiber Seca 306^®^ (Medical Measuring Systems and Scales SECA, Hamburg, Germany)with a maximum limit of 205 cm. All measurements were made in duplicate, with the participation of three certified anthropometrists, with Lin concordance rates that were higher than 0.99, following international protocols reported in the literature [17,18].

#### 2.2.2. Perivesicular Inflammation Findings

The presence of inflammation was determined: gallbladder wall thickening, necrosis and/or friable areas at the time of surgical intervention to integrate two groups: (1) patients without perivesicular inflammation and (2) patients with inflammation. Once the surgical procedure was completed, the findings were confirmed using histopathology.

#### 2.2.3. DPP-4 Serum Levels

Before the surgery was started, a venous blood sample was obtained after an eight-hour fast, which was collected in a Vacutainer tube^®^ (BD Vacutainer, México City, México) without EDTA. The sample was centrifuged at 3000 rpm for five minutes, collecting the serum to be frozen at −70 °C to the measurement of the serum levels of DPP-4. The determination of the levels of DPP-4 Serum was performed in duplicate in the Translational Research Laboratory in Pharmacology (LIFTAR), of the Faculty of Medicine of the Autonomous University of San Luis Potosí, through a commercial kit of ELISA HUMAN DPPIV/CD26 (#DY1180) (R&D System, Minneapolis, MN, USA), of R&D System with a *r* = 0.99, performing spectrophotometry reading to 480 nm and correction to 500 nm, following the manufacturer’s specifications.

### 2.3. Statistical Analysis

The continuous variables (age, body mass index BMI, percentage of body fat mass MG, and serum levels of DPP-4), were evaluated by the Shapiro-Wilks test and the quartile-quartile graphic test to determine their distribution.

Averages and standard deviation are presented for the normal and median variables and interquartile ranges for the non-normal ones. The variables sex, overweight/obesity and perivesicular inflammation are presented as frequencies and percentage. Serum DPP-4 levels among the formed groups were compared with the Mann-Whitney U test for non-parametric samples, while the rest of the variables were compared using the Student *t* test. The categorical variables were compared with the Chi square test. All data were analyzed in the statistical program RStudio version 1.1.453. (RStudio Inc., Boston, MA, USA), considering a value of *p* < 0.05 as statistically significant.

## 3. Results

### 3.1. Basic Information of the Participants

The prospective cohort from where these data was derived was developed in the period of May 2017 to November 2018, integrating a total of 120 patients scheduled to elective cholecystectomy of the General Surgery Service of the Central Hospital Dr. Ignacio Morones Prieto, who met the inclusion criteria mentioned and provided their informed consent. The patients were followed an average of 30 days post-surgery, and medical revisions were established. 88 patients who did not develop any complications (wound dehiscence, wound site infection, internal and respiratory infection) were integrated into this analysis.

90.9% of the sample was female, with an average age of 39.4 years, and an age range of 18–69 years. In the anthropometric variables we observed that at least more than 50% of our population had high values of BMI (mean 27.8 ± 64), abdominal circumference (94.7 ± 15.1) and percentage of body fat mass (34.7 ± 11.7), a situation that we see reflected with a combined frequency of overweight/Obese rate of 65.90% in accordance with the cutting points suggested by the World Health Organization (WHO). The average for DPP-4 serum levels was 3703.6 ± 1650.9. (Table 1).

Patients integrated into the study, did not present data on acute abdominal inflammatory processes prior to the procedure. In 27.3% of the intervened patients, intraoperative perivesicular inflammation was reported.

### 3.2. DPP-4 Serum Levels in Patients with Perivesicular Inflammation Data

From the trans-surgical findings of perivesicular inflammation, two groups were integrated: A group without perivesicular inflammation *n* = 64 and a group with inflammation findings *n* = 24. Among these groups there wasn’t significant differences between the following characteristics: Sex, age, BMI, abdominal circumference, percentage of fat mass and frequency of overweight/obesity (*p* = 0.154).

However, by comparing the serum levels of DPP-4, we observed that these were minor in the group of patients with data of perivesicular inflammation (3947.6 ± 1659.5 vs. 3053.2 ± 1469.6, LC 95% of the difference: 160.4–1628.3), being statistically significant (*p* = 0.018) (Table 2 and Figure 1).

## 4. Discussion

In the present study, the objective of comparing serum levels of DPP-4 was based on patients with surgical findings of perivesicular inflammation compared to controls, since as mentioned previously in the introduction, a finding of perivesicular inflammation that is not acute is difficult to determine for an inexperienced doctor and often can be masked even with the use of more specialized diagnostic tools like US or TAC.

The clinical features of the study’s participants that have been reported in this pathology are female patients that are older than 50 years and have a history of diabetes. These are factors that are frequently associated with complications, unlike those observed in our population, which included young women, with no history of diabetes [16].

One of the clinical features that stands out and is in accordance with what is reported in the literature is the presence of obesity in both groups. It should be mentioned that this group of patients had nutritional counseling to avoid excessive weight loss, which is another risk factor for the development of gallstones and complications [19].

Regarding the results observed in the serum levels of DPP-4, from the initial report as a possible biomarker in inflammatory processes or cancer, a great variability of results has been observed according to the pathology studied and its temporality. Decreases in periods of activity Rheumatoid Arthritis/Celiac disease and multiple Sclerosis have been seen, while increased activity have been observed in colorectal and esophageal cancers and more recently in thyroid pathologies. In cases of major depression and anorexia, the results are not conclusive, since extreme variations have been observed without relation to the activity [5]. Given these results, we decided to seek a population of study that was homogeneous, with the possibility of controlling for external factors such as diet, antibiotics, surgical management and frequency of laparoscopy.

In patients with Type 2 Diabetes Mellitus and obesity, increased levels of DPP-4 and also higher cardiovascular risk and insulin resistance were detected [20]. Unlike the aforementioned results, we observed low levels of DPP-4 in cases of perivesicular inflammation, although the combined frequency of overweight/obesity was almost 66%.

It is well known that chronic cholecystitis results from recurrent episodes of acute cholecystitis, and the causes of presentation may include cholelithiasis and episodes without stones and dyskinesia [19]. Recent studies such as Blackwood B et al. [21], showed that cases of chronic cholecystitis could be dismissed since results would depend on factors such as population study, diagnostic sensitivity of specialized studies, surgeon’s experience and multiple episodes of inflammation before cholecystectomy. However, it is worth mentioning that the limiting factor of this study in comparison with our results is the type of population, since they were pediatric patients. The latter led us to assume that inflammatory processes can be also underestimated in our population because inflammation do not always present in the same intensity, and patients do not always come back for revision or it’s difficult for patients to adhere to diet control and they may self-medicate without an appropriate record.

The results shown here may be due to the course of chronic cholecystitis with multiple exacerbations and recurrent episodes of acute inflammation associated to minimal pain, which can alter the secretion of DPP-4. As mentioned above, the response of the latter, given its pleyotrophic role, would be acting as an acute-phase reactant that could be of great diagnostic utility through early monitoring of its levels in this group of patients. However, given the cross-sectional design of the study, there was no post-operative control of DPP-4 levels, which could be the next step of this research.

The preoperative determination of the serum levels of DPP-4 could be oriented towards the possibility of finding chronic vesicular inflammation that was not previously visualized. This serum biomarker could increase the diagnostic precision and help the surgeon in the planning his surgical approach, with the purpose of avoiding major surgical complications or infectious processes. Future perspectives include the possibility of including patients with different abdominal pathologies and stages of activity, as well as different surgical approaches to corroborate the usefulness of this novel biomarker.

## 5. Conclusions

Our findings indicate a dual behavior of DPP-4 levels: in the presence of acute perivesicular inflammation, its metabolic pathway is activated in a similar way to that of an acute-phase reactant; in chronic phases there seems to be a consumption process that lowers the levels below those reported for subjects without inflammation. In this case, lower serum levels of DPP-4 could be used as an early biomarker of a severe and chronic inflammatory state. More studies are needed to help avoid the underestimation of chronic cholecystitis and unforeseen surgical complications.

## Figures and Tables

**Figure 1 medicina-55-00148-f001:**
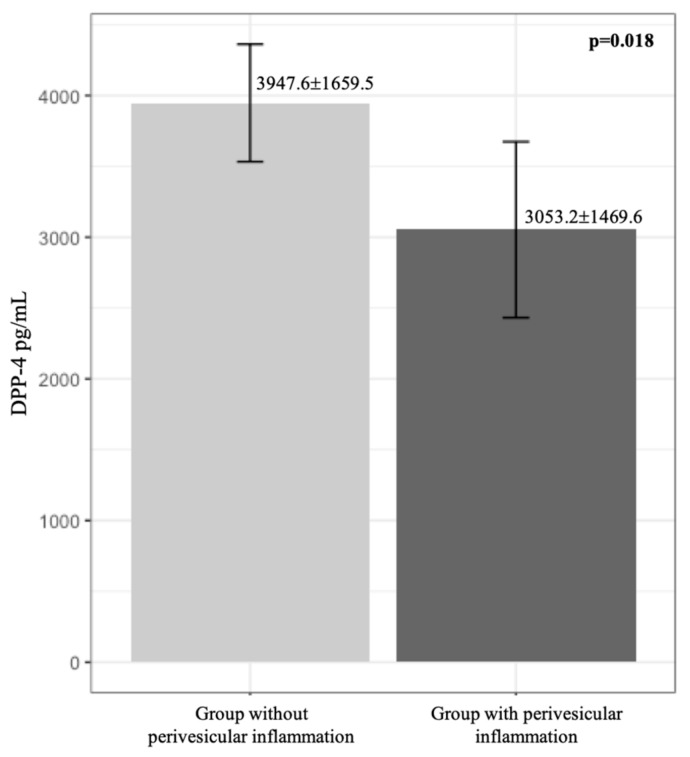
DPP-4 Serum levels in perivesicular inflammation. 3947.6 ± 1659.5 vs. 3053.2 ± 1469.6 pg/mL, *p* = 0.018.

**Table 1 medicina-55-00148-t001:** Basal characteristics of population.

Variable	*n* = 88
* Sex (Female)	90.90
* Age (Years)	39.4 ± 12.2
* Body Mass Index (kg/m^2^)	27.8 ± 6.4
^1^ OW/OB (%)	65.90
* Waist circumference (cm)	94.7 ± 15.1
** Percentage of body fat mass	34.7 ± 11.7
Finding of perivesicular inflammation (%)	27.3
* DPP-4 (pg/mL)	3703.6 ± 1650.9

* Mean and Standard Deviation. ** Median and interquartile range. ^1^ OW/OB: Overweight/obesity.

**Table 2 medicina-55-00148-t002:** Features of the groups with perivesicular inflammation findings and controls.

Variable	Group without Perivesicular Inflammation *n* = 64	Group with Perivesicular Inflammation *n* = 24	*p*-Value
° Sex (Female)	89.06	95.83	0.325
* Age (Years)	39.0 ± 11.7	40.6 ± 13.8	0.627
* Body Mass Index (kg/m^2^)	28.2 ± 5.3	26.7 ± 8.7	0.447
° OW/OB (%)	70.31	54.16	0.154
* Waist circumference (cm)	94.6 ± 14.1	94.8 ± 17.8	0.954
** Percentage of body fat mass	34.7 ± 10.1	31.5 ± 13.9	0.697
* DPP-4 (pg/mL)	3947.6 ± 1659.5	3053.2 ± 1469.6	**0.018**

* Mean and Standard Deviations, Student *t*-Test. ** Median and interquartile ranges, U test of Man-Whitney. ° Test Chi Square ^1^ OW/OB: Overweight/obesity.

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
