# Peer review of "DPP-4 as a Possible Biomarker of Inflammation Before Abdominal Surgery for Chronic Pathology: Our Experience with Elective Cholecystectomy"

_medicina, 2019, doi:10.3390/medicina55050148_

Round 1

Reviewer 1 Report

I understand that 120 patients were operated by elective cholecystectomy since May 2017 through Nov 2018.

A control group N = 64I u versus a group with perivesicular inflammation N = 24.

DPP-4 levels were reduced preoperatively in the group with perivesicular inflammation.

DPP-4 may help as an early biomarker of a severe and chronic inflammatory state.

DPP-4 helps diagnosing preoperative chronic cholecystitis.

At line 169, please change "perivascular" into "perivesicular".

Author Response

Point 1:I understand that 120 patients were operated by elective cholecystectomy since May 2017 through Nov 2018.  A control group N = 64I u versus a group with perivesicular inflammation N = 24. 

Response 1: The study is not a case control design; this issue has been commented 

 in the text, as suggested (page 3 line 99 to 112). 

Point 2:DPP-4 levels were reduced preoperatively in the group with perivesicular inflammation.

Response 1: this issue has been commented in the conclusion, as suggested (page 6 line 275 to 280). 

Point 3:DPP-4 may help as an early biomarker of a severe and chronic inflammatory state. 

Response 1: this issue has been commented in the conclusion, as suggested (page 6 line 275 to 280). 

Point 4: DPP-4 helps diagnosing preoperative chronic cholecystitis. 

Response 1: this issue has been commented in the conclusion, as suggested (page 6 line 275 to 280). 

Point 5:At line 169, please change "perivascular" into "perivesicular".

Response 1: we think that the grammatical corrector made the change "perivesicular" into "perivascular".This issue has been changed in the text. 

Reviewer 2 Report

In methods please keep separated inclusion and exclusion criteria

EG. We included parients between 18-70 years, with diagnosis of chronic cholecystitis  (within?) the 6 month before the inclusion, qualified as low cardio-metabolic risk in preoperative assessment, and those who were under dietary recommendations in the month prior to surgery.

History of peritonitis, pancreatitis, jaundice , previous ERCP, type 2 diabetes mellitus,

acute inflammatory (Protein C Reactive > 3mg/L, leucocytes >10 1000/mm3), neoplastic, nephrologic or liver disease, the use of anti-inflammatory drugs, steroids and/or antibiotics, presence of pacemakers or metallic implants, major amputations were all exclusion criteria.

All patients included agreed to participate by signing an informed consent.

Author Response

Point 1:In methods please keep separated inclusion and exclusion criteria

Response 1: this issue has been commented in methods as suggested (page 3, lines 99 to 112)

This manuscript is a resubmission of an earlier submission. The following is a list of the peer review reports and author responses from that submission.

Round 1

Reviewer 1 Report

Your conclusion: "with the latter,modulating a better immune response than could decrease the development of post-surgical complications" is not supported by the results of your study.

Please explain why low levels of DPP-4 in cases of perivesicular inflammation are important findings to be considered in medicine  and surgery.

Please change "perivascular" to "perivesicular" ( line 195 ).

Reviewer 2 Report

Title: please revise in order to comply with the topic of the research, eg. "DPP-4 as a possible biomarker of inflammation before abdominal surgery for subacute or chronic pathology: our experience with elective cholecistectomy"

Abstract: please report the inclusion and exclusion criteria. Specify that DPP-4 serum levels were measured preoperatively along with demographics (if possible standardize the timing of the measurement)

Methods:  

Participants: are patients who previously had jaundice and/or mild pancreatitis and who eventually underwent ERCP excluded?

Is advisable including in the analysis preoperative blood tests (white blood cell counts, C Reactive Proten, despite no significant differences are predicatable)

Furthermore, if possible, report postoperative blood tests (DPP-4 serum levels along with the abovementioned)

If possible standardize the timing of the measurements